

# Cornering extended Starobinsky inflation with CMB and SKA

Tanmoy Modak[1], Lennart Röver[1], Björn M. Schäfer[2],
Benedikt Schosser[1] and Tilman Plehn[1]

**1** Institut für Theoretische Physik, Universität Heidelberg, Germany
**2** Astronomisches Recheninstitut, Zentrum für Astronomie
der Universität Heidelberg, Germany

## Abstract

Starobinsky inflation is an attractive, fundamental model to explain the Planck measurements, and its higher-order extension may allow us to probe quantum gravity effects. We show that future CMB data combined with the 21cm intensity map from SKA will meaningfully probe such an extended Starobinsky model. A combined analysis will provide a precise measurement and intriguing insight into inflationary dynamics, even accounting for correlations with astrophysical parameters.


# 1   Introduction

Inflation [1–3] provides a simple and elegant solution to the observed flatness and horizon problems and naturally explains the absence of exotic relics. It also seeds primordial density fluctuations, from which the cosmic large-scale structure evolves. These structures can be observed in the cosmic microwave background (CMB) anisotropies [4, 5] and in the large-scale distribution of galaxies.

Among inflationary models, Starobinsky or $R^2$-inflation [1, 6–9] is one of the best-fitting models to data [5,10,11] of the early Universe. It simply extends the action of general relativity (GR) by a quadratic term in the Ricci-scalar. For the near-scale invariant power spectrum, deviations from GR manifest themselves primarily in a weak running of the spectral index. The value of the scalar amplitude and the spectral index reported by Planck [5, 10, 11] can be accounted for by adjusting the coefficient of the $R^2$-term. The extended Starobinsky model with higher-order curvature modifications is motivated by quantum gravity, but also from a purely phenomenological point of view [12–22] and it may shed light on the UV-completion of Einstein gravity. In this paper we extend the Starobinsky model by an $R^3$-term and study the constraining power of future cosmological data.

Planck's observations of the cosmic microwave background (CMB) temperature and polarisation anisotropies have advanced our understanding of inflation tremendously [5]. The next generation of CMB experiments will further develop this legacy. We focus on two future CMB experiments, LiteBIRD [23–25] and CMB-S4 [26–29]. The LiteBIRD satellite mission will detect primordial $B$-mode polarisation with moderate resolution, but excellent sensitivity. CMB-S4 stands for the next generation of ground-based detectors, which are going to be installed over the next decade, with excellent sensitivity and resolution, but limited sky coverage [26–29].

We supplement the CMB measurements with the 21cm intensity mapping by the Square Kilometre Array (SKA) [30–41], as a second window to primordial structures. We are primarily interested in the redshift range $z = 8 \dots 10$ and $k = 0.01 \dots 0.2$ Mpc$^{-1}$ [42]. The combined datasets well pick up variations in the spectral index to probe the extended Starobinsky model over a large range of scales. Structure formation at these scales is described well by linear physics with Gaussian statistics [43–46]. The low astrophysical systematics due to $X$-ray, UV-sources [47–51] or baryonic feedback processes [52–54] allow us to extract inflationary parameters from 21cm tomography. While we will use some simplifying assumptions, the modelling of the reionisation process at high redshift has reached a high degree of sophistication [55–60] and takes care of astrophysical processes, which are likewise modelled in machine learning approaches [61, 62].

In Sec. 2 we first discuss the details of the inflationary dynamics, deriving the required equivalent inflationary potential for extended Starobinsky models using the Einstein-Jordan duality. We then start with future CMB data and discuss the expected likelihoods for LiteBIRD and CMB-S4 in Sec. 3.1 and results in Sec. 3.2. In Sec. 4 we study the 21cm intensity mapping by SKA, again detailing the likelihood in Sec. 4.1, followed by a discussion of the modelling of the neutral hydrogen fraction as a function of redshift as the most important astrophysical parameter in Sec. 4.2. The results on probing the extended Starobinsky model with SKA and the next generation of CMB experiments are discussed in Sec. 4.3. We summarize our results in Sec. 5 and update our results on the slow-roll parametrization in the Appendix.

## 2 Extended Starobinsky model

The Starobinsky model [1, 6] is one of the simplest inflationary models, yet best-fitting to Planck data [5]. It is defined in the Jordan frame as

$$S_J = \frac{1}{2} \int d^4x \sqrt{-g_J} \, f(R),$$

(1)

where $g_J$ denotes the determinant of space-time metric $g_{\mu\nu_J}$ with signature convention $(-,+,+,+)$, $M_P = (8\pi G)^{-1/2}$, and

$$f(R) = M_P^2 \left( R + \frac{1}{6M^2} R^2 \right),$$

(2)

with $M^2 > 0$. The original Starobinsky model approximates general $f(R)$ gravity models with an attractor behavior in the large-field regime, where a single mass parameter $M$ accounts for the observed nearly-scale invariant power spectrum and spectral index [5]. Probing an actual inflationary potential complements results based on an effective reconstruction of inflationary potentials in the slow-roll approximation [42, 63]. We extend the original Starobinsky model by a $R^3$-curvature term,

$$f(R) = M_P^2 \left( R + \frac{1}{6M^2} R^2 + \frac{c}{36M^4} R^3 \right),$$

(3)

where $c$ is a dimensionless coefficient, which can be generated by quantum corrections. Higher-order terms involving derivatives, Ricci tensors and Riemann tensors typically involve ghosts [64], and we neglect them in favor of the $R^3$-term as a phenomenological window to physics beyond the simple Starobinsky model.

The corresponding scalar-tensor theory can be found by a Legendre transformation of Eq.(1),

$$S_J = \frac{1}{2} \int d^4x \sqrt{-g_J} \left[ f(s) + f'(s)(R - s) \right],$$

$$S_J \equiv \int d^4x \sqrt{-g_J} \left[ \frac{M_P^2}{2} \Omega^2 R - V(s) \right],$$

$$\text{with} \quad \Omega^2 = \frac{f'(s)}{M_P^2} = 1 + \frac{1}{3M^2} s + \frac{c}{12M^4} s^2$$

$$\text{and} \quad V(s) = \frac{1}{2} \left[ s f'(s) - f(s) \right].$$

(4)

The Legendre transform is well defined as long as $f(R)$ is convex, for Eq.(3) translating into $s > -2M^2/c$. The action in Eq.(1) can be expressed in the Einstein frame through the conformal transformation $g_{\mu\nu_E} = \Omega^2 g_{\mu\nu_J}$,

$$S_E = \int d^4x \sqrt{-g_E} \left[ \frac{M_P^2}{2} R_E - \frac{1}{2} g^{\mu\nu}_E \left( \nabla_\mu \varphi \nabla_\nu \varphi \right) - V_E(\varphi) \right],$$

(5)

with the canonical field $\varphi$ and

$$\varphi = \sqrt{\frac{3}{2}} M_P \ln \Omega^2,$$

(6)

$$V_E(\varphi) = \left. \frac{V(s)}{\Omega(s)^4} \right|_{s=s(\varphi)},$$

(7)

$$R = \Omega^2 \left[ R_E + 3 \Box_E \ln \Omega^2 - \frac{3}{2} g^{\mu\nu}_E \partial_\mu \ln \Omega^2 \, \partial_\nu \ln \Omega^2 \right].$$

(8)

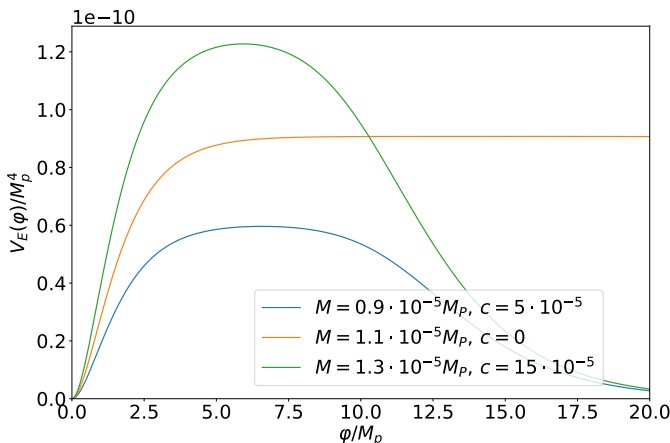

Figure 1: The shape of the inflationary potential for few reference choices of $M$ and $c$.

Here, $\Box_E = g_E^{\mu\nu}\partial_\mu\partial_\nu$ is the d'Alembert operator. This way, modifications of the gravitational law are mapped onto an additional field $\varphi$ subjected to dynamics in a potential $V(\varphi)$. This has the tremendous advantage that the standard inflationary formalism can be applied for computing the field dynamics and the associated generation of structures. In the potential one has to use $s(\varphi)$, as found by inverting $\Omega^2$ in Eq.(6) and solving for $s(\varphi)$. We find

$$
s(\varphi) = \begin{cases} \dfrac{2M^2}{c}\left[\sqrt{1 + 3c(e^{\sqrt{\frac{2}{3}}\frac{\varphi}{M_P}} - 1)} - 1\right], & \text{for } c \neq 0, \\[4mm] -3M^2\left[1 - e^{\sqrt{\frac{2}{3}}\frac{\varphi}{M_P}}\right], & \text{for } c = 0. \end{cases}
\tag{9}
$$

The potential can be expressed as

$$
V_E(\varphi) = \frac{M_P^2\left[\dfrac{cs(\varphi)^3}{M^2} + 3s(\varphi)^2\right]}{36M^2\left[1 + \dfrac{s(\varphi)}{3M^2} + \dfrac{cs(\varphi)^2}{12M^4}\right]^2}\,.
\tag{10}
$$

For $c = 0$ it can be put into the standard $R^2$ or Starobinsky form

$$
V_E(\varphi) = \frac{3M_P^2 M^2}{4}\left(1 - e^{-\sqrt{\frac{2}{3}}\frac{\varphi}{M_P}}\right)^2.
\tag{11}
$$

Here $s(\varphi)$ has two solutions, but from Eq.(9) we know that we need to satisfy the convexity condition $s > -M^2/(2c)$ and $c > 0$, while the potential $V_E(\varphi)$ has to remain positive at large field values. While the secondary solution can fulfill the convexity condition for $c < 0$, the potential becomes unbounded from below for large field values. In Fig 1 we illustrate $V_E(\varphi)$ for some sample parameter choices.

To study the inflationary dynamics we split $\varphi$ into a classical background $\bar{\varphi}$ and a perturbation $\delta\varphi$,

$$
\varphi(x^\mu) = \bar{\varphi}(t) + \delta\varphi(x^\mu)\,.
\tag{12}
$$

The perturbed spatially flat Friedmann-Robertson-Walker (FRW) metric can be expanded as [65–67]

$$ds^2 = -(1 + 2A)dt^2 + 2a(t)(\partial_i B)dx^i dt + a(t)^2 \left[(1 - 2\psi)\delta_{ij} + 2h_{ij}\right]dx^i dx^j, \tag{13}$$

where $a(t)$ is scale factor and $t$ is the cosmic time. The $A, B, \psi$ define scalar and $h_{ij}$ tensor metric perturbations.

With the above definitions the background field equation can be written as

$$\ddot{\bar{\varphi}} + 3H\dot{\bar{\varphi}} + V_{E,\bar{\varphi}} = 0, \tag{14}$$

where $H = \mathrm{d}(\ln a)/\mathrm{d}t$ is the Hubble function fulfilling

$$H^2 = \frac{1}{3M_P^2}\left[\frac{1}{2}\dot{\bar{\varphi}}^2 + V_E\right] \qquad \text{and} \qquad \dot{H} = -\frac{1}{2M_P^2}\dot{\bar{\varphi}}^2. \tag{15}$$

The slow-roll parameter $\epsilon$ can then be defined as

$$\epsilon \equiv -\frac{\dot{H}}{H^2}. \tag{16}$$

Inflation ends when $\epsilon = 1$.

Splitting $\varphi(x^\mu)$ into a background field $\bar{\varphi}(t)$ and gauge-dependent field fluctuations $\delta\varphi(x^\mu)$ motivates the gauge-independent Mukhanov-Sasaki variables for the fluctuations [66, 68–70],

$$Q = \mathcal{Q} + \frac{\dot{\bar{\varphi}}}{H}\psi, \qquad \text{with} \qquad \mathcal{Q} = D_\kappa\varphi|_{\kappa=0} = \frac{\mathrm{d}\varphi}{\mathrm{d}\kappa}|_{\kappa=0}, \tag{17}$$

where $\kappa$ is the trajectory in field space. The gauge-invariant field fluctuations $Q$ fulfill

$$\ddot{Q} + 3H\dot{Q} + \left[\frac{k^2}{a^2} + V_{E,\bar{\varphi}\bar{\varphi}} - \frac{1}{M_P^2 a^3}\frac{\mathrm{d}}{\mathrm{d}t}\left(\frac{a^3}{H}\dot{\bar{\varphi}}^2\right)\right]Q = 0, \tag{18}$$

where $V_{E,\bar{\varphi}\bar{\varphi}}$ is the double derivative of the potential $V_E(\bar{\varphi})$ with respect to $\bar{\varphi}$. The gauge-invariant curvature perturbation $\mathcal{R}$ is defined as [66, 67]

$$\mathcal{R} = \frac{H}{\dot{\bar{\varphi}}}Q, \tag{19}$$

and we are interested in the power spectrum of the gauge-invariant curvature perturbation [66, 71]

$$\langle \mathcal{R}(\boldsymbol{k}_1)\mathcal{R}(\boldsymbol{k}_2)\rangle = (2\pi)^3\delta_D^{(3)}(\boldsymbol{k}_1 + \boldsymbol{k}_2)P_\mathcal{R}(k_1), \qquad \text{with} \qquad P_\mathcal{R}(k) = |\mathcal{R}|^2. \tag{20}$$

The dimensionless power spectrum for the curvature perturbation is given by

$$\mathcal{P}_\mathcal{R}(t; k) = \frac{k^3}{2\pi^2}P_\mathcal{R}(k). \tag{21}$$

The spectral index $n_s$ of the power spectrum of the adiabatic fluctuations is defined as

$$n_s = 1 + \frac{\mathrm{d}\ln\mathcal{P}_\mathcal{R}(k)}{\mathrm{d}\ln k}. \tag{22}$$

On the other hand, the mode equation for the tensor amplitude is

$$v_{\boldsymbol{k}}'' + \left(k^2 - \frac{a''}{a}\right)v_{\boldsymbol{k}} = 0, \tag{23}$$

where $v_k$ is the gauge-invariant tensor amplitude and the prime denotes derivative with respect to conformal time $\tau$ defined by $dt = a\,d\tau$. The power spectrum of the tensor perturbations is expressed as

$$\mathcal{P}_{\mathcal{T}}(t;k) = 8\frac{k^3}{2\pi^2}|v_k|^2\,. \tag{24}$$

The tensor-to-scalar ratio $r$, *i.e.* the relative strength between the tensor and scalar power spectrum evaluated at reference scale $k_* = 0.05\ \mathrm{Mpc}^{-1}$, is defined as

$$r = \frac{\mathcal{P}_{\mathcal{T}}}{\mathcal{P}_{\mathcal{R}}}\,. \tag{25}$$

To determine the constraints on the Starobinsky model parameters $M$ and $c$ defined in Eq.(10) we solve the background and perturbation equations of Eq.(14), Eq.(18), and Eq.(23) in the Cosmic Linear Anisotropy Solving System (CLASS III) [72, 73].

# 3 Future CMB data

The first data we want to use to probe the inflationary potential are the CMB anisotropies, which probe the inflationary dynamics through their sensitivity to the structures in the early Universe. At the relevant redshifts around $z \simeq 10^3$ the cosmic large scale structure is to a very good approximation in a state of linear evolution. Additionally, the relationship between fluctuations in the gravitational potential, as predicted by linear perturbation theory, and the observable temperature and polarisation anisotropies is linear and is not tainted by astrophysics.

## 3.1 LiteBIRD and CMB-S4 likelihoods

While we will primarily focus on the future experiments LiteBIRD [23–25] and CMB-S4 [26–29], we also provide results based on Planck data [5] for validation. Going beyond Planck, future CMB measurements will improve the probe of small-scale fluctuations, allow better polarisation measurements, and address the $B$-mode polarisation as an imprint of tensor fluctuations on large scales. LiteBIRD mainly targets the large scale for polarisation but lacks sensitivity towards CMB lensing. On the other hand, CMB-S4 adds on this aspect significantly, except for large scales, where the small sky fraction and foreground due to lower sky coverage and fewer channels limits its reach [74].

We construct Gaussian likelihoods from all four possible spectra, $C_{TT}(\ell)$, $C_{TE}(\ell)$, $C_{EE}(\ell)$ and $C_{BB}(\ell)$. They are computed from the input spectra $\mathcal{P}_{\mathcal{R}}(k)$ and $\mathcal{P}_{\mathcal{T}}(k)$ which carry information about the inflationary potential given in Eq.(10), implemented in CLASS. Each CMB experiment is characterized by its sky fraction, its instrumental noise, and its angular resolution. They are incorporated into a covariance, for which we use a Gaussian approximation.

The gravitational lensing effect in the CMB smoothes out the spectra and, more importantly, converts between $E$-mode and $B$-mode polarisation. In our forecasts we assume the lensing effect to be modelled in the spectra, and we disregard the extracted deflection angle spectrum $C_{\alpha\alpha}(\ell)$ along with the cross-correlation $C_{\psi T}(\ell)$ between the lensing potential and the temperature fluctuation as a source of cosmological information. In light of the very strong signals from the primordial fluctuations, gravitational lensing would improve constraints on the background cosmology and the fluctuation amplitude marginally, but is not without risk, as the controversy about the Planck lensing amplitude demonstrated.

The evolution of the scalar and tensor perturbation spectra to the observable temperature and polarisation spectra of the CMB is handled by CLASS, and the resulting spectra are

assembled into a $\chi^2$-functional in a Markov Chain Monte Carlo (MCMC) framework MontePython3 [75,76]. A Markov chain generates samples from the likelihood $\mathcal{L} \propto \exp(-\chi^2/2)$ as a function of the fundamental cosmological parameters, along with the Starobinsky parameters $M$ and $c$. While we solve the mode equations for the Starobinsky model, we consider the subsequent evolution to be governed by standard general relativity. The mapping of the Starobinsky model from the Jordan to the Einstein frame makes the computations of the scalar and tensor spectra analogous to single-field inflation with a similar phenomenology of running spectral indices, so we can check our implementation against the standard $\alpha, \beta$-parametrization for $\mathcal{P}_\mathcal{R}(k)$.

We use the standard CMB-S4 and LiteBIRD likelihoods in MontePython, which are described in detail in Ref. [74]. For LiteBIRD the angular scales are $\ell = 2 \dots 1350$, the sky fraction is $f_{\mathrm{sky}} = 0.7$, while the channel is taken as 140 GHz with full-width-half-max or FWHM = 31 arcmin, $\Delta T = 4.1\ \mu$K arcmin, and $\Delta P = 5.8\ \mu$K arcmin. The CMB-S4 specifications are $\ell = 30 \dots 3000$, $f_{\mathrm{sky}} = 0.4$, 150 GHz channel, FWHM = 3 arcmin, $\Delta T = 1.0\ \mu$K arcmin and $\Delta P = 1.41\ \mu$K arcmin. We need to ensure that the two experiments cover mutually exclusive $\ell$ ranges, so just as in Ref. [74] we combine low-$\ell$ from LiteBIRD data and high-$\ell$ CMB-S4 data, separated at $\ell \le 50$. Noise is estimated through minimum variance estimator for both experiments. We use the HALOFIT [77] model for the nonlinear corrections throughout this paper.

## 3.2 Combined CMB projections

We use the combined estimated measurements from LiteBIRD and CMB-S4 to the fundamental parameters $M$ and $c$ in the extended Starobinsky potential. As the reference cosmological model we choose spatially flat $\Lambda$CDM-cosmology with parameter space spanned by $\{\omega_{\mathrm{b}}, \omega_{\mathrm{cdm}}, h, \tau_{\mathrm{reio}}\}$, along with the extended Starobinsky model parameters $\{M, c\}$, and $N_*$ as the number of $e$-foldings before the end of inflation, when the pivot scale $k_* = 0.05\ \mathrm{Mpc}^{-1}$ exits the horizon. We first consider Planck data, to see what the combined $TT, TE, EE$+low-$\ell EE$+low-$\ell TT$ spectra can tell about $M$ and $c$, with the baseline model parameters

$$\{ \omega_{\mathrm{b}}, \omega_{\mathrm{cdm}}, h, \tau_{\mathrm{reio}}, M, c, N_* \}. \tag{26}$$

For our MCMC runs in MontePython we use the Metropolis-Hastings algorithm and sample from a Gaussian proposal function with eight chains totaling up to 5.5 millions steps. We use flat priors for all parameters except for $N_*$, for which a Gaussian prior with mean $\mu_{N_*} = 55$ and standard deviation $\sigma_{N_*} = 5$ leads to a faster convergence of the chains. To check for convergence we use the criterion $R - 1 \lesssim 0.05$. The marginalized posterior distributions are shown by the green contours in Fig. 2 with the best fit, mean with errors and corresponding 95%CL limits given in Tab. 1. It is clear from the Tab. 1 that $c$ is compatible with zero, but showing a mild positive bias. Our marginalized values are completely compatible with Refs. [18,20,21].

Next, we take the best-fit values from the from Planck data shown in Fig. 2, specifically including

$$\frac{M}{M_P} = 1.103 \cdot 10^{-5} \qquad \text{and} \qquad c = 4.135 \cdot 10^{-5}, \tag{27}$$

and create likelihoods for LiteBIRD and CMB-S4, also discussed in the Appendix. Even though LiteBIRD and CMB-S4 are both CMB-experiments, their different focus on angular scales and polarisation renders them sensitive to cosmological parameters in different ways, as we see in Fig. 2. The baryon density $\omega_b$ is extracted from alternating peak heights of the acoustic peaks, so the large number of multipoles probed by CMB-S4 yields a better measurement of $\omega_b$. A similar argument applies to the matter density $\omega_{\mathrm{cdm}}$, reflected in the sequence of higher order peaks, where again CMB-S4 has an advantage. For inflation parameters $M$ and

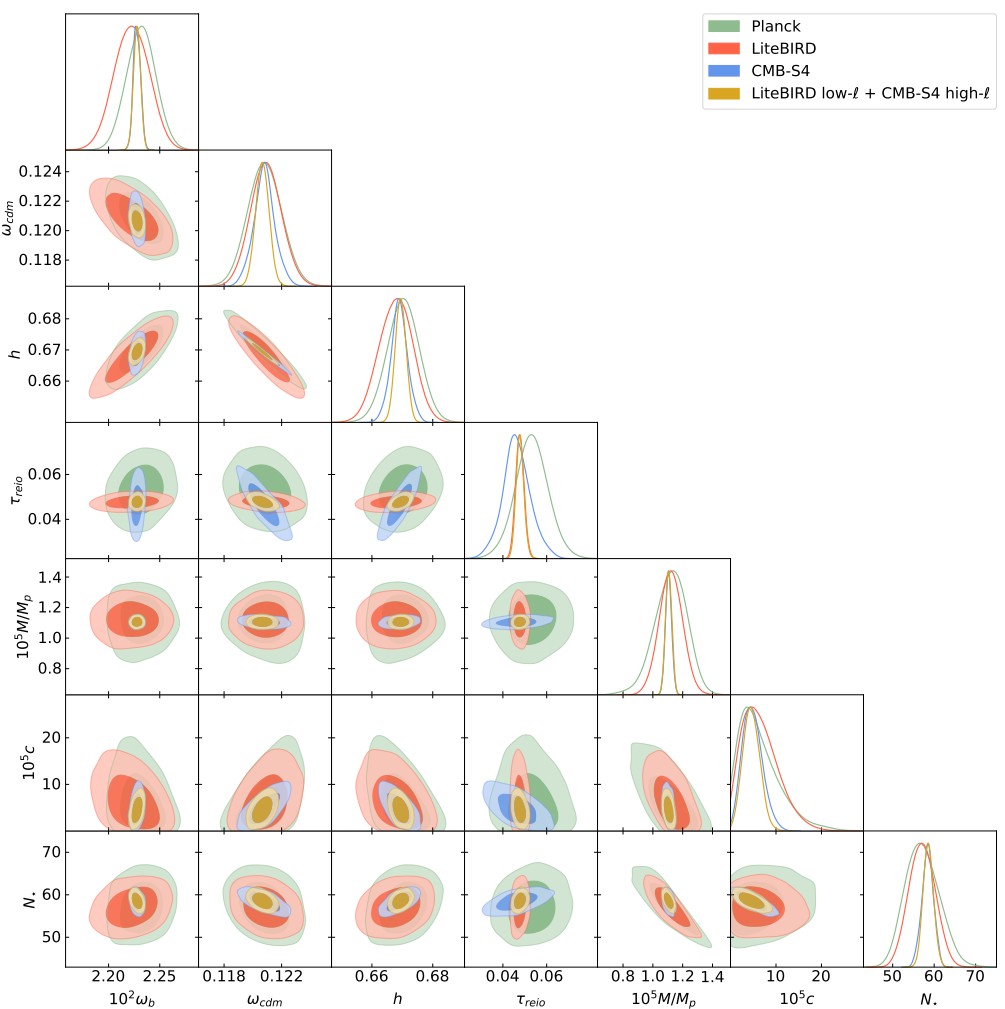

Figure 2: Marginalized CMB posteriors for the extended Starobinsky model, based on Planck ($TT$, $TE$, $EE$+low-$\ell EE$+low-$\ell TT$), LiteBIRD, CMB-S4, and the consistent combination of LiteBIRD and CMB-S4.

$c$, the much larger $\ell$-values probed by CMB-S4 can also be seen to make a difference. In contrast, measuring the optical depth $\tau_{\rm reio}$ requires excellent polarisation sensitivity on large scales, giving LiteBIRD a clear advantage. Still, the results and especially the control over the astrophysics nuisance parameters of the inflation measurement improves significantly when we combined LiteBIRD low-$\ell$ with CMB-S4 high-$\ell$ data, allowing us to measure the assumed value $c = 4.135 \cdot 10^{-5}$ to

$$c = (1.015 \ldots 8.3) \cdot 10^{-5} \qquad (95\% \mathrm{CL}). \tag{28}$$

We briefly remark that adding lensing data to $TT$, $TE$, $EE$+low-$\ell EE$+low-$\ell TT$ only provides minor improvements, which we do not show. While gravitational lensing of the CMB is included in our modelling, we do not carry out a lensing reconstruction, which yields the deflection angle spectra and the cross-correlation between the lensing potential and the temperature map [78]. Although CMB-lensing is a source of cosmological information, it is a resource-intensive analysis with moderate improvements on inflationary constraints. Controlling the lensing-induced mode conversion between $EE$ and $BB$ is well-investigated in the literature, and these results also applies to the Starobinsky case with running spectral indices [79].

Table 1: Best-fit values, mean, error bars, and 95%CL limits for the parameters shown in Fig. 2.

| Data | Parameters | Best-fit | Mean±$\sigma$ | 95% lower | 95% upper |
|---|---|---|---|---|---|
| Planck $(TT, TE, EE$+low-$\ell EE$ +low-$\ell TT$) | $100\,\omega_b$ | 2.228 | $2.232^{+0.015}_{-0.015}$ | 2.203 | 2.26 |
| | $\omega_{\rm cdm}$ | 0.1206 | $0.1208^{+0.0012}_{-0.0012}$ | 0.1185 | 0.1232 |
| | $h$ | 0.6696 | $0.6703^{+0.0053}_{-0.0053}$ | 0.6600 | 0.6808 |
| | $\tau_{\rm reio}$ | 0.04781 | $0.05315^{+0.0074}_{-0.0077}$ | 0.03764 | 0.0687 |
| | $10^5 M/M_P$ | 1.103 | $1.119^{+0.117}_{-0.0987}$ | 0.9005 | 1.329 |
| | $10^5 c$ | 4.135 | $6.069^{+2.840}_{-5.402}$ | — | < 15.96 |
| | $N_\star$ | 58.24 | $57.17^{+3.73}_{-4.47}$ | 49.65 | 65.24 |
| LiteBIRD | $100\,\omega_b$ | 2.229 | $2.223^{+0.018}_{-0.017}$ | 2.190 | 2.256 |
| | $\omega_{\rm cdm}$ | 0.1204 | $0.1209^{+0.001}_{-0.0011}$ | 0.1188 | 0.1231 |
| | $h$ | 0.6705 | $0.6679^{+0.0057}_{-0.0055}$ | 0.657 | 0.6785 |
| | $\tau_{\rm reio}$ | 0.04735 | $0.04775^{+0.002}_{-0.002}$ | 0.04391 | 0.05171 |
| | $10^5 M/M_P$ | 1.144 | $1.121^{+0.077}_{-0.077}$ | 0.9676 | 1.273 |
| | $10^5 c$ | 2.633 | $6.345^{+2.996}_{-4.801}$ | — | < 14.62 |
| | $N_*$ | 57.79 | $57.08^{+3.18}_{-3.19}$ | 51.04 | 63.27 |
| CMB-S4 | $100\,\omega_b$ | 2.227 | $2.228^{+0.004}_{-0.004}$ | 2.221 | 2.235 |
| | $\omega_{\rm cdm}$ | 0.121 | $0.1208^{+0.0007}_{-0.0007}$ | 0.1192 | 0.1223 |
| | $h$ | 0.6681 | $0.669^{+0.0027}_{-0.0027}$ | 0.6634 | 0.6749 |
| | $\tau_{\rm reio}$ | 0.04478 | $0.04634^{+0.0064}_{-0.0058}$ | 0.03258 | 0.05963 |
| | $10^5 M/M_P$ | 1.098 | $1.105^{+0.021}_{-0.021}$ | 1.065 | 1.145 |
| | $10^5 c$ | 5.166 | $4.794^{+1.923}_{-2.461}$ | 0.7769 | 9.543 |
| | $N_*$ | 58.44 | $58.45^{+1.45}_{-1.35}$ | 55.66 | 61.26 |
| LiteBIRD low-$\ell$ + CMB-S4 high-$\ell$ | $100\,\omega_b$ | 2.227 | $2.228^{+0.004}_{-0.004}$ | 2.221 | 2.235 |
| | $\omega_{\rm cdm}$ | 0.1206 | $0.1207^{+0.0005}_{-0.0005}$ | 0.1197 | 0.1216 |
| | $h$ | 0.6696 | $0.6695^{+0.0018}_{-0.0018}$ | 0.6659 | 0.673 |
| | $\tau_{\rm reio}$ | 0.04829 | $0.04779^{+0.0017}_{-0.0019}$ | 0.04425 | 0.05148 |
| | $10^5 M/M_P$ | 1.108 | $1.106^{+0.022}_{-0.021}$ | 1.064 | 1.147 |
| | $10^5 c$ | 4.177 | $4.573^{+1.786}_{-1.944}$ | 1.015 | 8.300 |
| | $N_*$ | 58.71 | $58.59^{+1.24}_{-1.25}$ | 56.15 | 61.08 |

# 4 SKA data

As a second probe of inflationary dynamics we focus on fluctuations in the 21cm background generated by spin-flip transitions of neutral hydrogen. The 21cm background is generated at much lower redshifts around $z \lesssim 10$. This implies that, depending on the redshift window considered, nonlinearities could become important on small scales. Intricacies of reionising radiation sources, radiative transport, and details of the reionising process would then limit our analysis. We target $z = 8 \dots 10$ and employ a simplified model to compute fluctuations in the 21cm intensity from the statistics of the matter distribution with weak non-linearities described by the halo-model.

## 4.1 SKA likelihood

As outlined in Sec. 3.1, we evolve the spectra of the scalar and tensor perturbations with CLASS, and in parallel to the CMB-spectra we compute the density perturbation spectrum

$P_\delta(k)$ to model the 21cm-intensity spectrum. The 21cm-spectra depend on the wave number $k$, the orientation of the modes relative to the line of sight $\mu$, and the redshift $z$. They are assembled into a tomographic, redshift-resolved measurement for maximising the sensitivity. The likelihood is a $\chi^2$-functional, constructed assuming a Gaussian covariance with the experimental characteristics of SKA. It can be combined with CMB-likelihoods, assuming statistical independence. Here, a caveat are the integrated Sachs-Wolfe and the gravitational lensing effects in the CMB, which are generated by foreground structures that are directly mapped by their 21cm emission, introducing a weak correlation [80].

We incorporate details of the 21cm emission through a redshift-dependent bias parameter as well as a factor taking care of redshift space distortions induced by peculiar velocities. We model the reionisation history with a simple 2-parameter model that captures the global properties of the reionisation process and is verified against 21cmFAST [81, 82].

We follow closely Ref. [83] for the evaluation of 21cm power spectrum in our target redshift range. Assuming a flat-sky approximation [84, 85], the Fourier mode $\vec{k}$ and the line-of-sight $\vec{r}$ describe the power spectrum in terms of

$$
k = \left| \vec{k} \right| \qquad \text{and} \qquad \mu = \frac{\vec{k} \cdot \vec{r}}{kr} \, , \tag{29}
$$

with the $k$-components $k_\perp = k\sqrt{1-\mu^2}$ and $k_\parallel = \mu k$. This gives us

$$
P_{21}(k,\mu,z) = f_{\mathrm{AP}}(z) \times f_{\mathrm{res}}(k,\mu,z) \times f_{\mathrm{RSD}}(\hat{k},\hat{\mu},z) \times b_{21}^2(z) \times P_\delta(\hat{k},z) \, . \tag{30}
$$

The wave-number $k$ and the orientation of a mode relative to the line of sight $\mu$ are derived quantities, as one needs for a given redshift the angular diameter distance and the Hubble-function which themselves depend on cosmology. Therefore, it is necessary to differentiate between the values $k$ and $\mu$ in the cosmological model probed in our analysis from the assumed-truth or fiducial parameters describing the assumed cosmology $\hat{k}$ and $\hat{\mu}$. $P_\delta$ is the matter power spectrum from CDM and baryons and

$$
b_{21} = \overline{\Delta T_b}(z) b_{\mathrm{HI}}(z), \quad \text{with} \quad \overline{\Delta T_b} \simeq 189 \left[ \frac{H_0 \, (1+z)^2}{H(z)} \right] \Omega_{\mathrm{HI}}(z) \, h \, \mathrm{mK}, \tag{31}
$$

with the mean differential brightness temperature expressed in terms of the reduced Hubble parameter $h$ defined through $H_0 = h \times 100 \, \mathrm{km/(s \; MPc)}$. In addition, $b_{\mathrm{HI}}(z)$ is an, in principle, redshift-dependent bias. For simplicity we neglect the redshift dependence in $b_{\mathrm{HI}}$ and treat it as a nuisance parameter. The mass density of neutral hydrogen with respect to critical density is given by

$$
\Omega_{\mathrm{HI}}(z) = \frac{\rho_{\mathrm{HI}}}{\rho_c} = \Omega_b (1 - Y_P) \left( \frac{H_0}{H(z)} \right)^2 (1+z)^3 \, x_{\mathrm{HI}}(z), \tag{32}
$$

with $\Omega_b = 0.0495$. $Y_P = 0.24672$ [4] is the primordial helium fraction, and $x_{\mathrm{HI}}(z)$ is the neutral hydrogen fraction discussed in detail in Sec. 4.2.

Going back to Eq.(30), the so-called Alcock-Paczinsky effect, or the relative change in the power spectrum between true and the assumed true (i.e. fiducial) cosmology, is accounted for by

$$
f_{\mathrm{AP}}(z) = \frac{D_A^2 \hat{H}}{\hat{D}_A^2 H} \, , \tag{33}
$$

where $H$ and $D$ are the Hubble parameter and angular diameter distance as a function of $z$. Quantities within the true cosmology are denoted with $\hat{\phantom{x}}$, e.g. $\hat{H}$. The Fourier-modes are characterised by wave number $k$ and orientation $\mu$ relative to the line of sight, where the

relation in these quantities between the true cosmology and and assumed cosmological model is given by

$$
\begin{aligned}
\hat{k}^2 &= \left[\frac{\hat{H}}{H}^2 \mu^2 + \frac{D_A}{\hat{D}_A}(1-\mu^2)\right]k^2\,, \\
\hat{\mu}^2 &= \frac{\hat{H}}{H}^2 \mu^2 \left[\frac{\hat{H}}{H}^2 \mu^2 + \frac{D_A}{\hat{D}_A}(1-\mu^2)\right]^{-1}\,.
\end{aligned}
\tag{34}
$$

Next, $f_{\mathrm{res}}(k,\mu,z)$ describes the finite resolution of the instruments, which suppresses the perturbations on small scales,

$$
f_{\mathrm{res}}(k,\mu,z) = \exp\left[-k^2\left(\mu^2(\sigma_\parallel^2 - \sigma_\perp^2) + \sigma_\perp^2\right)\right]\,,
\tag{35}
$$

where $\sigma_\parallel$ and $\sigma_\perp$ are the Gaussian errors of the coordinates parallel and perpendicular to the line of sight at redshift $z$. They are given by

$$
\sigma_\parallel = \frac{c}{H}(1+z)^2\frac{\sigma_\nu}{\nu_0} \qquad \text{and} \qquad \sigma_\perp = (1+z)D_A\sigma_\theta\,,
\tag{36}
$$

$$
\text{with} \qquad \sigma_\theta = \frac{1}{\sqrt{8\ln 2}}\frac{\lambda_0}{D_{\mathrm{base}}}(1+z)z \quad \text{and} \quad \sigma_\nu = \frac{\delta_\nu}{\sqrt{8\ln 2}}\,.
\tag{37}
$$

The first quantity is the Gaussian suppression of the power spectrum defined as the ratio between the root mean square and a FWHM of $\sqrt{8\ln 2}$. The latter corresponds to the channel width due to the band separation into different channels with $\lambda_0 = 21.11$ cm, which translates to $\nu_0 = 1420.405752$ MHz. We use the SKA1-LOW specifications [39], expected for observing in one band $\nu = 50 \dots 350$ MHz, where the 21cm line in our target redshift $z = 8 \dots 10$ lies. The core SKA1-LOW configuration is an array of 224 antennas with diameter $D = 40$ m and with maximum baseline $D_{\mathrm{base}} = 1$ km [39]. Here, we use 64000 channels [86] with $D_{\mathrm{base}} = 1$ km, again for SKA1-LOW [39].

Finally, the classical cosmological redshift induces an apparent anisotropy in the power spectrum, as described by the Kaiser formula [87] in the linear regime. Furthermore, the random peculiar velocities of the galaxies lead to the so-called fingers-of-God effect [88] in the redshift. Both are included through the term $f_{\mathrm{RSD}}$ Eq.(30) and described by [89]

$$
f_{\mathrm{RSD}}(\hat{k},\hat{\mu},z) = \left(1 + \beta(\hat{k},z)\hat{\mu}^2\right)^2 e^{-\hat{k}^2\hat{\mu}^2\sigma_{\mathrm{NL}}^2}\,, \qquad \text{with} \qquad \beta(\hat{k},z) = -\frac{1+z}{2b_{21}(z)}\frac{\mathrm{d}\log P_\delta(\hat{k},z)}{\mathrm{d}z}\,.
\tag{38}
$$

This form of $\beta$ is valid for $k = 0.01 \dots 0.2$ Mpc$^{-1}$ and $z = 8 \dots 10$. The first term represents the Kaiser formula, the exponential term the fingers of God. We take $\sigma_{\mathrm{NL}} = 1$ Mpc as our fiducial value, which corresponds to non-linear scale of $k_{\mathrm{NL}} = 1$ Mpc$^{-1}$. Due to our conservative $k$-range, this choice has very little effect.

The entire observed 21cm power spectrum is a combination of the signal and noise [90],

$$
P_{21}^{\mathrm{obs}}(k,\mu,z) = P_{21}(k,\mu,z) + P_N(z)\,, \qquad \text{with} \qquad P_N(z) = \frac{4\pi T_{\mathrm{sys}}^2 f_{\mathrm{sky}}\lambda^2 y D_A^2}{A\Omega f_{\mathrm{cover}}t_{\mathrm{obs}}}\,.
\tag{39}
$$

Here $t_{\mathrm{obs}}$ is the total observation time which we take to be 10000 hrs, $N_{\mathrm{dish}}$ is the number of antennas, $f_{\mathrm{sky}} = 0.58$. In our analysis we consider a field of view of $\Omega = (1.2\lambda/D)^2$, an area $A = N_{\mathrm{dish}}\pi(D/2)^2$ per antenna, and the covering fraction $f_{\mathrm{cover}} = N_{\mathrm{dish}}(D/D_{\mathrm{base}})^2$. Again, we follow the design specification of SKA1-LOW [39]. The system temperature is the combination of the sky temperature and the receiver temperature [39]

$$
\begin{aligned}
T_{\mathrm{sys}} &= T_{\mathrm{sky}} + T_{\mathrm{rx}}\,, \\
\text{with} \quad T_{\mathrm{sky}} &= 25\,\mathrm{K}\left(\frac{408\,\mathrm{MHz}}{\nu}\right)^{2.75} \quad \text{and} \quad T_{\mathrm{rx}} = 0.1T_{\mathrm{sky}} + 40\,\mathrm{K}\,,
\end{aligned}
\tag{40}
$$

and $\nu = \nu_0/(1 + z)$. Unlike Ref. [83], where the noise model treats SKA as a single-dish experiment, our noise model is based on interferometry. Furthermore, $y$ is defined as

$$y = \frac{18.5\text{MPc}}{1\,\text{MHz}} \left( \frac{1 + z}{10} \right)^{1/2}. \tag{41}$$

For the 21cm intensity mapping, we divide the mapping into bins of width $\Delta z$ with mean redshift $\bar{z}$. The volume of one redshift bin can then be approximated as

$$V_r(\bar{z}) = 4\pi f_{\text{sky}} \int_{\Delta r(\bar{z})} r^2 \mathrm{d}r = \frac{4\pi}{3} f_{\text{sky}} \left[ r^3 \left( \bar{z} + \frac{\Delta z}{2} \right) - r^3 \left( \bar{z} - \frac{\Delta z}{2} \right) \right]. \tag{42}$$

The Gaussian $\chi^2$ giving the likelihood is then defined as the integral over $k$ and $\mu$ for each redshift band as [83]

$$\chi^2 = \sum_{\text{bins } n} \int_{k_{\text{min}}}^{k_{\text{max}}} k^2 \mathrm{d}k \int_{-1}^{1} \mathrm{d}\mu \frac{V_r(\bar{z}_n)}{2(2\pi)^2} \left[ \frac{(\Delta P_{21}(k, \mu, \bar{z}_n))^2}{(P_{21}(k, \mu, \bar{z}_n) + P_N)^2 + \sigma_{\text{th}}^2(k, \mu, \bar{z}_n)} \right], \tag{43}$$

where $\Delta P_{21}$ is the difference between the fiducial and sampled power spectra, and

$$\sigma_{\text{th}}(k, \mu, z) = \left[ \frac{V_r(z)}{2(2\pi)^2} k^2 \Delta\mu \Delta k \frac{\Delta z}{\Delta \bar{z}} \right]^{1/2} \alpha(k, \mu, z) P_{21}(k, \mu, z). \tag{44}$$

This uncertainty depends on the correlation lengths $(\Delta k, \Delta\mu, \Delta z)$. For a given bin $(k_i, z_j)$, the choice of $\Delta\mu$ depends on the number of independent nuisance parameters describing the errors for different $\mu_k$. Following Ref. [83], for a given bin $(k_i, z_j)$ the error on $P_{21}(k, \mu, z)$ for different $\mu$ values can be treated as fully correlated. Taking one parameter per bin is then equivalent to $\Delta\mu = \mu_{\text{max}} - \mu_{\text{min}} \approx 2$ for our redshift range, reducing Eq.(44) to

$$\sigma_{\text{th}}(k, \mu, z) = \left[ \frac{V_r(z)}{(2\pi)^2} k^2 \Delta k \frac{\Delta z}{\Delta \bar{z}} \right]^{1/2} \alpha(k, \mu, z) P_{21}(k, \mu, z). \tag{45}$$

The correlation length $\Delta k$ is assumed to be 0.05 $h/\text{Mpc}$ as a conservative choice, matching the BAO scale. We also choose $\Delta z = 1$, which is slightly lower than the whole redshift range probed by the experiment $z_{\text{max}} - z_{\text{min}} = 2$.

The function $\alpha(k, \mu, z)$ accounts for three uncertainties from different non-linear corrections: The prediction of the matter power spectrum, the bias, and RSD. Even though non-linear effects are small in our target redshift range, we include them in our analysis, except for the RSD source which is negligible for $z = 8 \dots 10$. The bias is usually assumed to be linear up to scales $k = 0.2\,h/\text{Mpc}$. The HALOFIT semi-analytic formula, which we use, includes some of these effects, but not baryonic and AGN feedback. To account for the corresponding uncertainties in the bias and RSD at small scales we increase the theoretical uncertainties for three reference points [83], to a 0.33% error at $k = 0.01\,h/\text{Mpc}$, a 1% error at $k = 0.3\,h/\text{Mpc}$, and a 3% error at $k = 10\,h/\text{Mpc}$. This translates into

$$\alpha(k, z) = \begin{cases} a_1 \exp\left( c_1 \log_{10} \frac{k}{k_1(z)} \right), & \text{for } \frac{k}{k_1(z)} < 0.3, \\ a_2 \exp\left( c_2 \log_{10} \frac{k}{k_1(z)} \right), & \text{for } \frac{k}{k_1(z)} > 0.3, \end{cases}$$
$$k_1(z) = 1 \frac{h}{\text{Mpc}} (1 + z)^{\frac{2}{2+n_s}}, \tag{46}$$

with $a_1 = 1.4806\%$, $a_2 = 2.2047\%$, $c_1 = 0.75056$, and $c_2 = 1.5120$. As a conservative implementation we apply a sharp cut-off at $k = 0.2\,h/\text{Mpc}$ following the $z$-dependent scaling of Eq.(46).

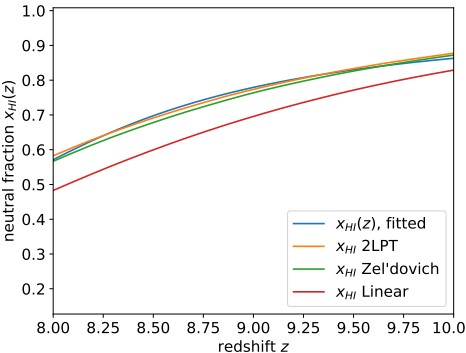 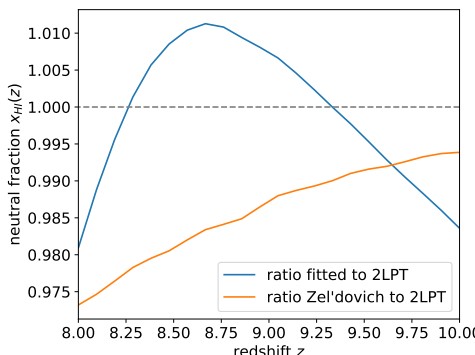

Figure 3: Evolution of the average neutral hydrogen fraction with redshift. The fitting function in Eq.(47) yields a precision comparable to the Zel'Dovich approximation.

The SKA likelihood is, again, implemented in MontePython, with a fiducial likelihood based on the best-fit values Planck ($TT$, $TE$, $EE$+low-$\ell EE$+low-$\ell TT$) shown in Tab. 1. Our updated power spectrum includes effects which were not considered in our earlier study [42], such as the linear biasing factor, the redshift dependence of neutral hydrogen fraction, $f_{AP}(z)$, and $f_{res}$. The noise model is also significantly improved by considering the realistic specifications of SKA in the high redshift region [39]. As astrophysical inputs for the 21cm power spectrum we focus on the reionisation history, modelled by the reionisation redshift, and the velocity at which the Universe transitions from being neutral to being ionised. Our modeling is tested against radiative transfer simulations in Gaussian random fields with 21cmFAST, confirming that it captures the relevant physics.

## 4.2 Modeling the redshift dependence

To describe the $z$-dependence of $x_{HI}$ in Eq.(32) we use the empirical fitting formula

$$x_{HI}(z) = \frac{1}{2}\left[1 + \frac{2}{\pi}\tan^{-1}\left(\delta_1(z - \delta_2)\right)\right], \qquad (47)$$

where $\delta_1$ and $\delta_2$ are again nuisance parameters. The functional shape of Eq.(47) is chosen to fit simulated data from 21cmFAST [81,82]. In our target redshift region $z = 8 \dots 10$, the neutral hydrogen fraction is extracted using the default parameters of 21cmFAST. For each of the 22 linearly spaced redshift bins a cube with side lengths 200 Mpc is simulated in real space. The computation is carried out on a $300 \times 300 \times 300$ grid, using the default astrophysics settings of 21cmFAST. The initial power spectrum is chosen to match CLASS, which corresponds to the cosmological parameters $\omega_b = 0.02237$, $\omega_{cdm} = 0.120$, $h = 0.6736$, $A_s = 2.100 \cdot 10^{-9}$, $n_s = 0.9649$, and $z_{reio} = 11.357$. These parameters are computed from the ones defined in Eq.(26).

For each of the simulated cubes we compute the average neutral hydrogen fraction using a first-order perturbative approximation (Zel'dovich's approximation) and a second-order 2LPT approximation to the linear velocity field in 21cmFAST. We then perform a one-dimensional fit of Eq.(47) to the 2LPT results, giving $\delta_1 = 0.9755$ and $\delta_2 = 7.7664$ as fiducial values for our MCMC runs. Fig. 3 illustrates the quality of this approximation and shows that the relative difference between Eq.(47) and the 2LPT result is at most 2% in our redshift region of interest.

## 4.3 Combined SKA and CMB projections

We now turn our attention to the sensitivity of SKA to the extended Starobinsky model parameters. Without any CMB information the SKA power spectrum is not sufficient to constrain all

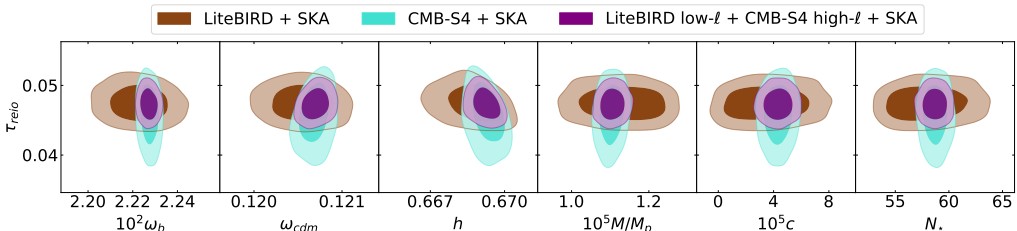

Figure 4: Correlation between $\tau_{\mathrm{reio}}$ and the remaining parameters of Eq.(26), based on LiteBIRD, CMB-S4, and the combination of LiteBIRD low-$\ell$ and CMB-S4 high-$\ell$ with SKA.

parameters given in Eq.(26). However, a combination with the Planck data is already sufficient to provide a convincing measurement [42]. Here we ask the more challenging question, namely what does SKA add to the combination of LiteBIRD and CMB-S4. In Fig. 4 we compare the combined sensitivity of SKA with LiteBIRD, CMB-S4 and their low-$\ell$ and high-$\ell$ combination, respectively. We only show the correlations of $\tau_{\mathrm{reio}}$ to the remaining parameters, where we see the excellent polarisation sensitivity on large scales from LiteBIRD. For all other parameters there is no additional constraining power from LiteBIRD and the contours are dominated by CMB-S4.

In Fig. 5 we compare the combined sensitivity of LiteBIRD low-$\ell$, CMB-S4 high-$\ell$, and SKA with the CMB sensitivity alone. The corresponding best-fit, mean and corresponding 95%CL limits are given in Tab. 2. The astrophysical parameters benefiting significantly from SKA are $\omega_{\mathrm{cdm}}$ and $h$. While we are mainly interested in the fundamental parameters of the inflation potential, this kind of improvement leads to a big improvement in the global analysis. While the combination with SKA still leaves a narrow correlation between the astrophysical $N_*$ and the Starobinsky parameter $M$, it provides an improved reach in the second Starobinsky parameter, as compared to the CMB projection of Eq.(28),

$$c = (2.89 \ldots 5.73) \cdot 10^{-5} \qquad (95\%\mathrm{CL}) \, . \tag{48}$$

The narrow correlations between $N_*$ and $M$ and, to some extent, $c$ trace back to how Eq.(14) is solved. The initial conditions to solve Eq.(14) in CLASS require the number of $e$-foldings before the end of inflation when the reference mode exited the horizon *i.e.* $N_*$ and the magnitude of $M$ and $c$. This solution is then used to match the observables $A_s$ and $n_s$, and leads to the strong correlation found above. Such a correlation can perhaps be resolved with the better description of the (p)reheating process after inflation.

# 5 Outlook

We have estimated the sensitivity of future CMB and SKA measurements to the Starobinsky model for inflation, extended by a $R^3$-term. Such a term may hint at physics beyond general relativity, including quantum gravity. Planck data prefers a finite $R^2$-terms and constrains the coefficient of the $R^3$-term to be $c \lesssim 1.6 \times 10^{-4}$ at 95%CL.

We performed a global Markov chain analysis, combining astrophysical and cosmological parameters with the two fundamental parameters describing Starobinsky inflation. First, we found that future CMB data from LiteBIRD and CMB-S4 will constrain the astrophysical parameters and also the inflationary parameters $M$ and $c$. In particular, we found that combining the two experiments in mutually exclusive $\ell$ ranges can probe the coefficient of $R^3$ at the level $c = (1.01 \ldots 8.3) \times 10^{-5}$ at 95%CL. The assumed finite central value is given by the best-fit value from our Planck analysis.

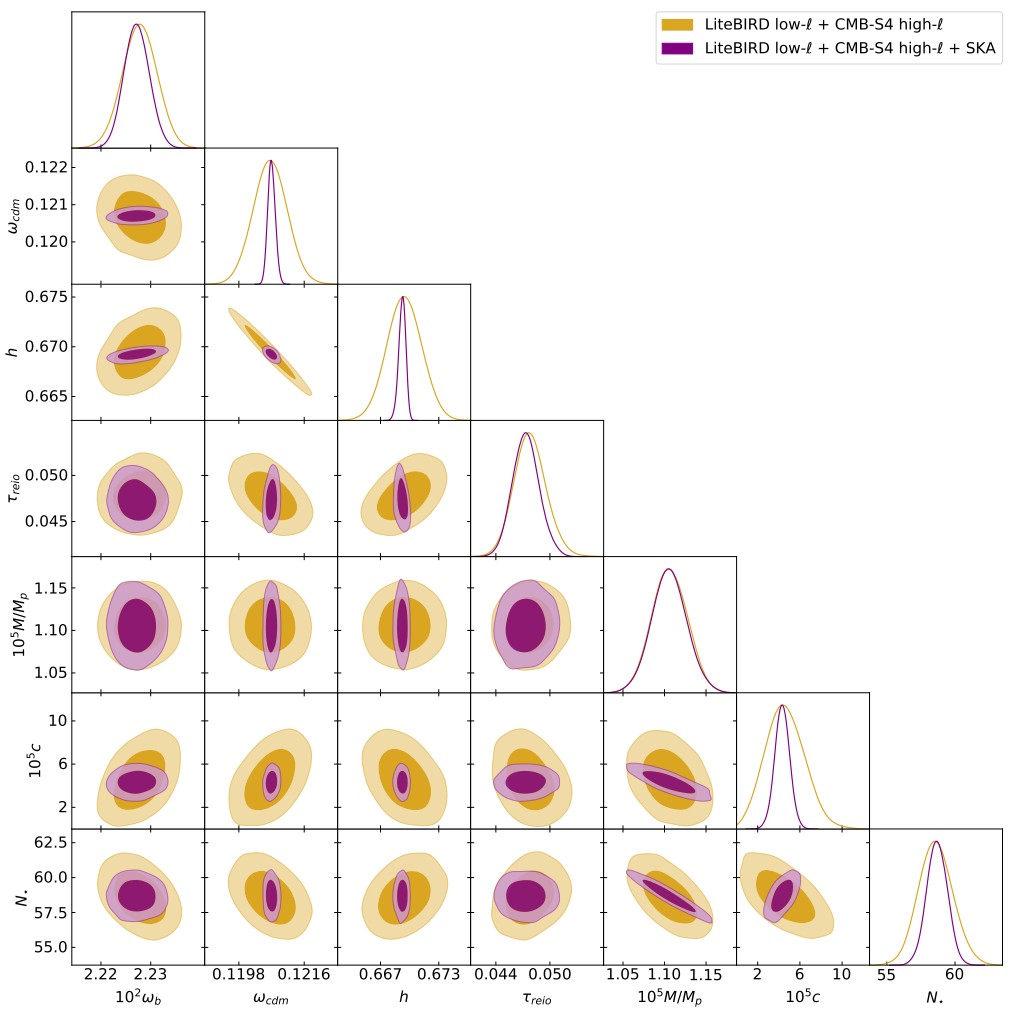

Figure 5: Marginalized CMB and SKA posteriors for the extended Starobinsky model, based on the combination of LiteBIRD and CMB-S4 with high-$\ell$+SKA projections.

Next, we showed that 21cm intensity mapping by SKA will add to the constraints from CMB data, focusing on the redshift region $z = 8 \dots 10$. While the combination of future CMB and SKA data still leaves us with a sizeable correlation between the number of $e$-foldings $N_*$ and

Table 2: Best-fit values, mean, error bars, and 95%CL limits for the parameters shown in Fig. 5.

| Data | Parameters | Best-fit | Mean±$\sigma$ | 95% lower | 95% upper |
|---|---|---|---|---|---|
| LiteBIRD low-$\ell$ + CMB-S4 high-$\ell$ + SKA | $100\,\omega_b$ | 2.228 | $2.227^{+0.003}_{-0.003}$ | 2.222 | 2.232 |
| | $\omega_{\mathrm{cdm}}$ | 0.1206 | $0.1207^{+0.0001}_{-0.0001}$ | 0.1205 | 0.1209 |
| | $h$ | 0.6694 | $0.6692^{+0.0004}_{-0.0003}$ | 0.6685 | 0.670 |
| | $\tau_{\mathrm{reio}}$ | 0.04792 | $0.04734^{+0.0014}_{-0.0016}$ | 0.04445 | 0.05033 |
| | $10^5 M/M_P$ | 1.100 | $1.106^{+0.023}_{-0.023}$ | 1.064 | 1.148 |
| | $10^5 c$ | 4.350 | $4.325^{+0.692}_{-0.690}$ | 2.891 | 5.734 |
| | $N_*$ | 58.95 | $58.68^{+0.77}_{-0.75}$ | 57.20 | 60.18 |

the scalaron mass $M$, it improves the measurement of the extended Starobinsky parameters to $c = (2.9 \ldots 5.7) \times 10^{-5}$. If $c$ is non-zero, SKA will allow for a robust determination of this fundamental parameter pointing to physics beyond standard GR.

# Acknowledgments

TM thanks Sung Mook Lee, Kin-ya Oda, and Tomo Takahashi for fruitful discussions.

**Funding information** TM is supported by Postdoctoral Research Fellowship from Alexander von Humboldt Foundation. The research of TP is supported by the Deutsche Forschungsgemeinschaft (DFG, German Research Foundation) under grant 396021762 - TRR 257 Particle Physics Phenomenology after the Higgs Discovery. This work was supported by the Deutsche Forschungsgemeinschaft (DFG, German Research Foundation) under Germany's Excellence Strategy EXC 2181/1 - 390900948 (the Heidelberg STRUCTURES Excellence Cluster).

# A HSR projections

Finally, we briefly revisit our previous results on the combination of Planck and SKA [42] and determine the the sensitivity of future CMB data in terms of the so-called Hubble slow-roll (HSR) parameters [5,63]. In this parametrization, the inflationary dynamics are captured by reconstructing the Hubble function in the observable window, defined by the range of observationally accessible spatial scales as

$$H(\varphi) = \sum_{n=0}^{N} \frac{1}{n!} \left. \frac{\mathrm{d}^n H}{\mathrm{d}\varphi^n} \right|_{\bar{\varphi}_*} (\bar{\varphi} - \bar{\varphi}_*)^n \,. \tag{A.1}$$

To avoid degeneracies it is convenient to use the logarithmic changes to the Hubble function through the parameters [5,63]

$$\lambda_H^{(n)} = \left( \frac{m_{\mathrm{Pl}}^2}{4\pi} \right)^n \left( \frac{(H')^{n-1}}{H^n} \frac{\mathrm{d}^{n+1}H}{\mathrm{d}\varphi^{n+1}} \right) \,, \qquad n \geq 1 \,, \tag{A.2}$$

with the correspondence $\eta_H = \lambda^{(1)}$, $\xi_H^2 = \lambda^{(2)}$, and $\omega_H^3 = \lambda^{(3)}$. As in before, we assume spatially flat $\Lambda$CDM-cosmology with the baseline model, as described by $\{\omega_{\mathrm{b}}, \omega_{\mathrm{cdm}}, h, \tau_{\mathrm{reio}}, n_s, \tilde{A}_s, \epsilon_H, \eta_H, \xi_H^2, \omega_H^3\}$, where we truncate the HSRs after $\omega_H^3$. The purpose of this Appendix is to investigate the power of future CMB data in constraining HSRs, along with a new SKA likelihood with an improved signal and noise modeling as compared to Ref. [42].

To determine the projected constraints on the HSR parameters we rely on CLASS and MontePython, as discussed in the main body of the paper. The expected constraints are shown for Planck, LiteBIRD, CMB-S4, and Planck+SKA in Fig 6 while, for LiteBIRD low-$\ell$+CMB-S4 high-$\ell$ and LiteBIRD low-$\ell$+CMB-S4 high-$\ell$+SKA they are shown in Fig.7. The respective best-fit and mean values are given in Tab. 3. As in the extended Starobinsky model, both LiteBIRD+CMB-S4 and LiteBIRD+CMB-S4+SKA data will provide the best constraints. We note that the fiducial likelihoods for the LiteBIRD, CMB-S4 and SKA are generated with the best-fit values to the marginalized posterior of Planck $TT$, $TE$, $EE$+low-$\ell$ $EE$+low-$\ell$ $TT$ data, also given in Tab. 3.

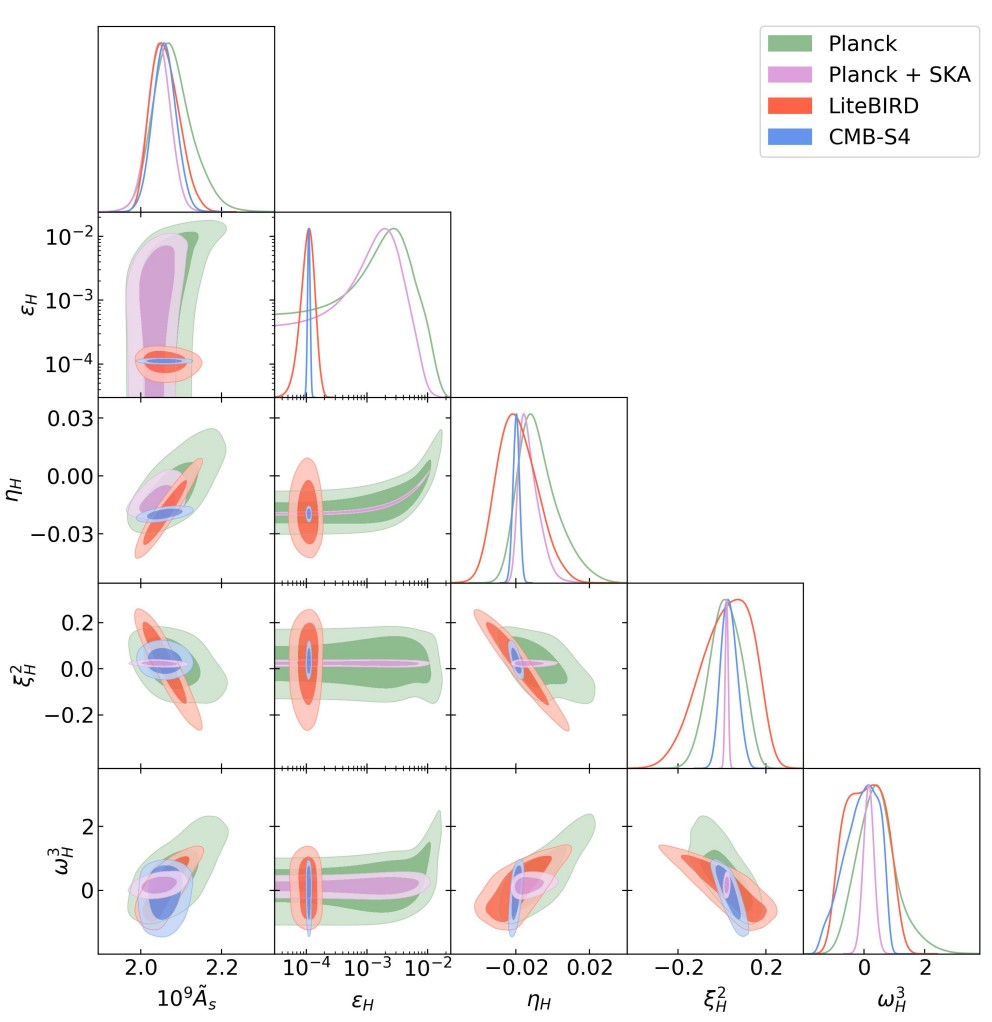

Figure 6: Marginalized CMB posteriors for the HSR parameters based on Planck ($TT$, $TE$, $EE$+low-$\ell EE$+low-$\ell TT$), Planck+SKA, LiteBIRD, CMB-S4.

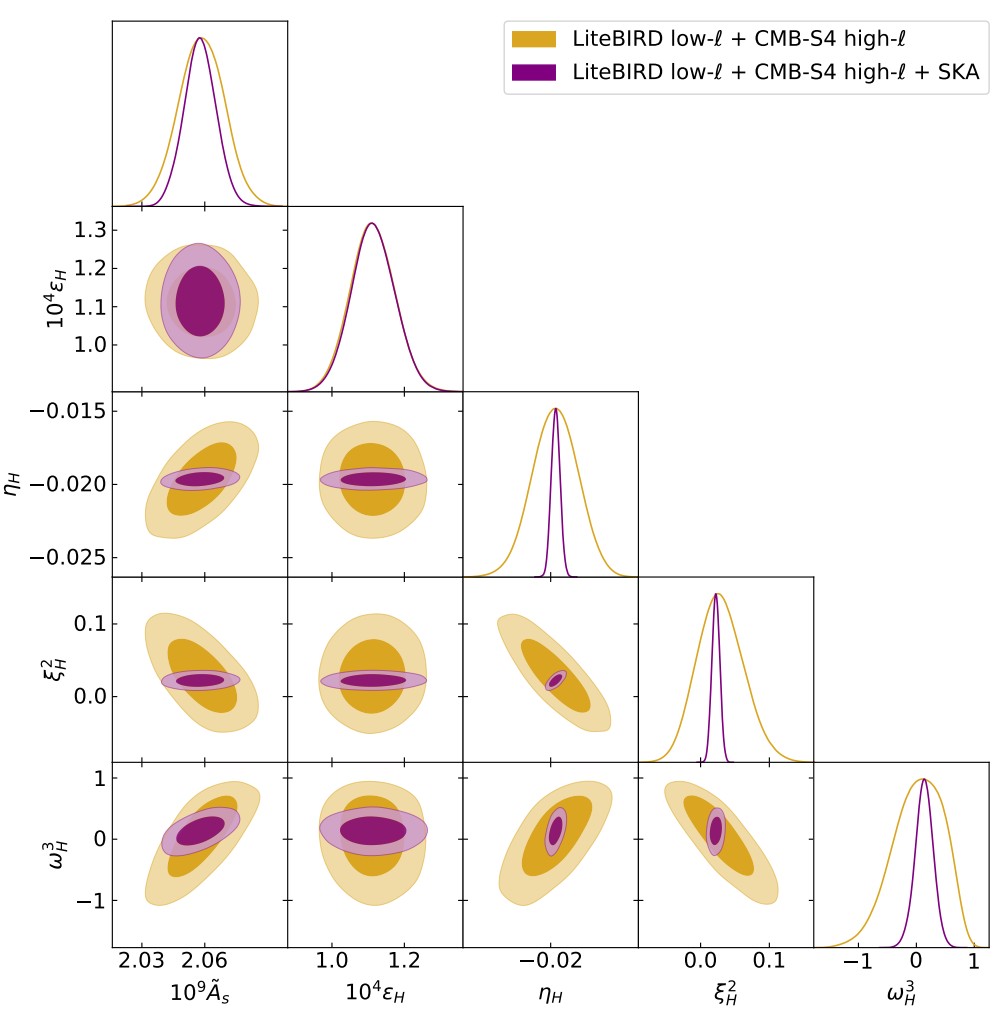

Figure 7: Marginalized CMB posteriors for the HSR parameters based on LiteBIRD low-$\ell$+CMB-S4 high-$\ell$ and LiteBIRD low-$\ell$+CMB-S4 high-$\ell$+SKA.

Table 3: Best-fit values, mean, error bars, and 95%CL limits for the HSR parameters shown in Figs. 6 and 7.

| Data | Parameters | Best-fit | Mean$\pm\sigma$ | 95% lower | 95% upper |
|---|---|---|---|---|---|
| Planck | $10^9\tilde{A}_s$ | 2.059 | $2.080^{+0.039}_{-0.057}$ | 1.987 | 2.186 |
| | $\epsilon_H$ | 0.0001111 | $0.005445^{+0.002930}_{-0.005313}$ | — | < 0.01393 |
| | $\eta_H$ | −0.01953 | $-0.007755^{+0.007628}_{-0.012983}$ | −0.02712 | 0.01599 |
| | $\xi^2_H$ | 0.02493 | $0.01809^{+0.07614}_{-0.07272}$ | −0.1175 | 0.1587 |
| | $\omega^3_H$ | 0.1008 | $0.4431^{+0.5342}_{-0.7216}$ | −0.8662 | 1.812 |
| Planck + SKA | $10^9\tilde{A}_s$ | 2.034 | $2.047^{+0.028}_{-0.029}$ | 1.986 | 2.102 |
| | $\epsilon_H$ | 0.001220 | $0.003338^{+0.001678}_{-0.003156}$ | — | < 0.009013 |
| | $\eta_H$ | −0.0175 | $-0.01283^{+0.00353}_{-0.00658}$ | −0.02158 | −0.001015 |
| | $\xi^2_H$ | 0.02386 | $0.02238^{+0.00687}_{-0.00676}$ | 0.00856 | 0.0356 |
| | $\omega^3_H$ | 0.01487 | $0.1594^{+0.5060}_{-0.1722}$ | −0.1846 | 0.5023 |
| LiteBIRD | $10^9\tilde{A}_s$ | 2.052 | $2.061^{+0.030}_{-0.042}$ | 1.995 | 2.131 |
| | $10^4\epsilon_H$ | 1.086 | $1.151^{+0.260}_{-0.316}$ | 0.5912 | 1.736 |
| | $\eta_H$ | −0.01991 | $-0.01897^{+0.00938}_{-0.01229}$ | −0.03931 | 0.002793 |
| | $\xi^2_H$ | 0.04889 | $0.02338^{+0.13927}_{-0.09424}$ | −0.2021 | 0.2296 |
| | $\omega^3_H$ | −0.2849 | $0.04961^{+0.67578}_{-0.68982}$ | −1.059 | 1.155 |
| CMB-S4 | $10^9\tilde{A}_s$ | 2.06 | $2.059^{+0.028}_{-0.030}$ | 2.002 | 2.118 |
| | $10^4\epsilon_H$ | 1.104 | $1.113^{+0.053}_{-0.057}$ | 1.007 | 1.221 |
| | $\eta_H$ | −0.0191 | $-0.0198^{+0.0018}_{-0.0018}$ | −0.02334 | −0.01632 |
| | $\xi^2_H$ | 0.01098 | $0.03418^{+0.03331}_{-0.03942}$ | −0.03461 | 0.1062 |
| | $\omega^3_H$ | 0.3613 | $-0.09232^{+0.73948}_{-0.37747}$ | −1.157 | 0.8509 |
| LiteBIRD low-$\ell$ + CMB-S4 high-$\ell$ | $10^9\tilde{A}_s$ | 2.053 | $2.059^{+0.011}_{-0.011}$ | 2.037 | 2.08 |
| | $10^4\epsilon_H$ | 1.109 | $1.112^{+0.060}_{-0.064}$ | 0.9925 | 1.235 |
| | $\eta_H$ | −0.01999 | $-0.01966^{+0.00166}_{-0.00165}$ | −0.02291 | −0.01636 |
| | $\xi^2_H$ | 0.02682 | $0.0288^{+0.0316}_{-0.0359}$ | −0.03628 | 0.09762 |
| | $\omega^3_H$ | −0.003998 | $0.03371^{+0.51047}_{-0.37005}$ | −0.7958 | 0.8296 |
| LiteBIRD low-$\ell$ + CMB-S4 high-$\ell$ + SKA | $10^9\tilde{A}_s$ | 2.06 | $2.058^{+0.008}_{-0.008}$ | 2.042 | 2.073 |
| | $10^4\epsilon_H$ | 1.114 | $1.114^{+0.060}_{-0.062}$ | 0.9967 | 1.235 |
| | $\eta_H$ | −0.01951 | $-0.01964^{+0.00031}_{-0.00033}$ | −0.02026 | −0.01902 |
| | $\xi^2_H$ | 0.02687 | $0.02238^{+0.00568}_{-0.00568}$ | 0.01128 | 0.03339 |
| | $\omega^3_H$ | 0.1544 | $0.1347^{+0.1561}_{-0.1525}$ | −0.1974 | 0.4407 |

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
