# Peer review of "Cornering Extended Starobinsky Inflation with CMB and SKA"

_SciPost Physics, doi:SciPost Phys. 15, 047 (2023)_

## Round 1 · Referee Report · Lucien Heurtier · 2023-2-24

Strengths

1- Important new direction of cosmological searches
2- Clarity of the paper

Weaknesses

1- The results are quite model dependent, but that is unavoidable

Report

Dear editor,

The authors propose to constrain Starobinsky inflation models, supplemented by $R^3$ terms using future CMB observatories such as LiteBird and CMB-S4, and 21 cm intensity mapping experiments such as the square kilometre array SKA. Indeed, by focusing on redshifts and scales over which structure formation is known to be under control, and where the expected background is reduced, the authors hope to constrain inflation parameters using 21 cm line measurements.

After reviewing basic results about inflation and its treatment in perturbation theory, the authors describe how they obtain likelihoods for LiteBird and CMB-S4 and derive best-fit values for the set of cosmological and inflation parameters considered. Then the authors explain how they derive the 21 cm line power spectrum in the redshift range they consider, and obtain the corresponding SKA likelihood. Using the combined sensitivities of the three detectors, the authors demonstrate that the three detectors can help measure better the value of the extra term they add to the action.

The manuscript is clear and well-written. The purpose of the paper is modest but well-motivated. I believe it deserves to be published in SciPost Physics, as it opens a new window on searches for signatures of inflation (and modified gravity), not only using CMB observations but also other classes of cosmological measurements such as 21cm line measurements.

However, before the paper is accepted for publication, I would need the authors to address the following questions/comments:

1) Although it is understandable in the text by deduction that it corresponds to fiducial values, the parameter $\hat H$ and $\hat D$ used in Eq.(33) are not defined.

2) In Eq.(37), the choice of $\sigma_{\rm NL} = 1$Mpc as a fiducial value should be justified. How would a different choice affect the results?

3) After Eq.(44), it is mentioned that the authors "choose $\Delta z = 1$, which is slightly lower than the whole redshift range probed by the experiment zmax − zmin = 2." The authors should comment on the motivation and validity for such a choice.

4) The quoted best-fit value from Planck for the c-parameter (times 1e5) is 4.135 in Eq.(27) and before Eq.(28). However, in Table 1, the best-fit value obtained from Planck is 4.315. The authors should make sure of which value is the correct one, and possibly cross-check that the numbers in the Tables do not contain any other typos if there is one.

5) An open question: The authors notice in their analysis a correlation between the value of (M,c) and the value of the number of e-folds of inflation N_star. Such a correlation is expected, as constraints on inflation are inherently entangled with the assumed post-inflationary cosmic history. In this work the authors set the prior for N_star to be a gaussian centred on 55 with width 5, to cover a range of 50-60 e-folds. First I am wondering whether such choice for the prior affects significantly the final results. Second, while pure radiation domination from the end of inflation to the later Big-Bang cosmology would lead to a large value of N_star, a long period of early matter domination could lower this number even below 50. While this early cosmic history may not frankly affect CMB measurements beyond the single value of N_star, I am wondering whether it could have an impact on the 21cm intensity mapping as the matter power spectrum may evolve differently in the presence of a long period of early matter domination. Also, I am wondering whether CLASS assumes a specific cosmic history (maybe pure radiation domination right after inflation?), in which sense it may not be clear what it means to actually vary the number of e-folds of inflation without modifying cosmic history.

Best regards
The referee

  • validity: good
  • significance: good
  • originality: high
  • clarity: high
  • formatting: perfect
  • grammar: good

Author:  Benedikt Schosser  on 2023-03-24  [id 3504]

(in reply to Report 1 by Lucien Heurtier on 2023-02-24)

Dear referee,

Thank you for your questions and remarks. Our answers are compactly attached in a file. Additionally, you can find a new version of the manuscript with the changes in red attached.

Best regards,
The editor

Attachment:

answers_report.pdf

Anonymous on 2023-04-06  [id 3559]

(in reply to Benedikt Schosser on 2023-03-24 [id 3504])

Dear Editor,

I am happy with the answers provided by the authors and the amendment made to the draft.

I am thus now considering the manuscript acceptable for publication in SciPost.

Best regards
The referee

---

## Editorial Decision

published